# Elevated Levels of Circulating Hsp70 and an Increased Prevalence of CD94+/CD69+ NK Cells Is Predictive for Advanced Stage Non-Small Cell Lung Cancer

**DOI:** 10.3390/cancers14225701

**Published:** 2022-11-21

**Authors:** Sophie Seier, Ali Bashiri Dezfouli, Philipp Lennartz, Alan Graham Pockley, Henriette Klein, Gabriele Multhoff

**Affiliations:** 1Department of Radiation Oncology, Central Institute for Translational Cancer Research Technische Universität München (TranslaTUM), Klinikum Rechts der Isar, 81675 Munich, Germany; 2John van Geest Cancer Research Centre, School of Science and Technology, Nottingham Trent University, Nottingham NG11 8NS, UK; 3multimmune GmbH, 81675 Munich, Germany; 4Division of Thoracic Surgery, Klinikum Rechts der Isar, TUM, 81675 Munich, Germany

**Keywords:** circulating heat-shock protein 70, tumor biomarker, non-small-cell lung cancer (NSCLC), UICC stage, CD4+ T cell drop, liquid biopsy

## Abstract

**Simple Summary:**

This study reveals that circulating Hsp70 levels could serve as a tumor biomarker for patients with NSCLC in advanced UICC stages. All patients in advanced tumor stages had significantly elevated Hsp70 levels in the circulation compared to a healthy control cohort and an early-stage tumor cohort, and Hsp70 levels progressively increased with higher UICC tumor stages. These findings demonstrate the potential of Hsp70 measurements to predict an advanced tumor stage in NSCLC patients. We have also demonstrated that the prevalence of CD3−/CD94+ NK cells and CD8+ cytotoxic T cells were greater in advanced tumor stages, whereas that of CD4+ T helper cells was decreased. We hypothesize that raised levels of circulating Hsp70 in higher tumor stages might support NK cell proliferation, but that a lowered prevalence of CD4+ T helper cells could temper the capacity of cytolytic CD8+ T cells and NK cells to control tumor growth.

**Abstract:**

Non-small cell lung cancer (NSCLC) is the second most frequently diagnosed tumor worldwide. Despite the clinical progress which has been achieved by multimodal therapies, including radiochemotherapy, and immune checkpoint inhibitor blockade, the overall survival of patients with advanced-stage NSCLC remains poor, with less than 16 months. It is well established that many aggressive tumor entities, including NSCLC, overexpress the major stress-inducible heat shock protein 70 (Hsp70) in the cytosol, present it on the plasma membrane in a tumor-specific manner, and release Hsp70 into circulation. Although high Hsp70 levels are associated with tumor aggressiveness and therapy resistance, membrane-bound Hsp70 can serve as a tumor-specific antigen for Hsp70-primed natural killer (NK) cells, expressing the C-type lectin receptor CD94, which is part of the activator receptor complex CD94/NKG2C. Therefore, we investigated circulating Hsp70 levels and changes in the composition of peripheral blood lymphocyte subsets as potential biomarkers for the advanced Union for International Cancer Control (UICC) stages in NSCLC. As expected, circulating Hsp70 levels were significantly higher in NSCLC patients compared to the healthy controls, as well as in patients with advanced UICC stages compared to those in UICC stage I. Smoking status did not influence the circulating Hsp70 levels significantly. Concomitantly, the proportions of CD4+ T helper cells were lower compared to the healthy controls and stage I tumor patients, whereas that of CD8+ cytotoxic T cells was progressively higher. The prevalence of CD3−/CD56+, CD3−/NKp30, CD3−/NKp46+, and CD3−/NKG2D+ NK cells was higher in stage IV/IIIB of the disease than in stage IIIA but were not statistically different from that in healthy individuals. However, the proportion of NK cells expressing CD94 and the activation/exhaustion marker CD69 significantly increased in higher tumor stages compared with stage I and the healthy controls. We speculate that although elevated circulating Hsp70 levels might promote the prevalence of CD94+ NK cells in patients with advanced-stage NSCLC, the cytolytic activity of these NK cells also failed to control tumor growth due to insufficient support by pro-inflammatory cytokines from CD4+ T helper cells. This hypothesis is supported by a comparative multiplex cytokine analysis of the blood in lung cancer patients with a low proportion of CD4+ T cells, a high proportion of NK cells, and high Hsp70 levels versus patients with a high proportion of CD4+ T cells exhibiting lower IL-2, IL-4, IL-6, IFN-γ, granzyme B levels.

## 1. Introduction

After breast cancer, lung cancer is the second most frequently diagnosed tumor disease worldwide and, with 1.8 million deaths, is the most common reason for cancer-related mortality [1]. Compared to other tumor entities, no standard screening methods for the detection of lung cancer are available. Symptoms, such as coughing, dyspnea, chest pain, tiredness, and weight loss, are unspecific and occur also in smokers or patients with chronic obstructive pulmonary disease (COPD). Therefore, lung cancer is often diagnosed in an advanced tumor stage, which is associated with poor clinical outcomes. Advanced-stage non-small cell lung cancer (NSCLC) has been treated for a long time with platinum-based chemotherapy and ionizing irradiation, either sequentially or simultaneously [2]. Recent studies have demonstrated a significant benefit in overall survival in a small subgroup of NSCLC patients who additionally received an immune checkpoint inhibitor blockade, such as the PD-L1 inhibitor Durvalumab [3]. However, a relevant proportion of patients still do not profit from this immunological approach, and therefore, progression-free survival in patients with high-grade NSCLC remains less than 16 months [4]. One reason for this disappointing clinical outcome is the lack of reliable biomarkers for the early detection and prediction of response to an immune checkpoint inhibitor blockade in NSCLC. 

The highly conserved, stress-inducible heat shock protein 70 (Hsp70) could serve as a promising candidate for a universal tumor-specific biomarker. A cytosolic overexpression of Hsp70 has been detected in multiple solid tumor entities, such as the lungs [5], colorectal [6], prostate [7] carcinoma, and glioblastoma [8], as well as malignant hematological diseases. Under physiological conditions, Hsp70 acts as a molecular chaperone for protein maturation and degradation during cell growth and differentiation. In nearly all tumor cell types, the synthesis of Hsp70 is highly upregulated due to the cellular stress caused by enhanced glucose metabolism and faster proliferation rates [9]. In addition to the cytosolic overexpression, tumor cells present Hsp70 on their plasma membrane. This membrane localization of Hsp70 is enabled by a tumor-specific sphingolipid composition which enables the anchorage of Hsp70 in the plasma membrane [10]. Membrane Hsp70 on tumor cells acts as a “double-edged sword”; on the one hand, a high membrane Hsp70 density is associated with increased tumor aggressiveness [11], therapy resistance [12], and an elevated likelihood of metastatic spread [13]. On the other hand, membrane-bound Hsp70 serves as a target for NK cells [14] that have been pre-stimulated with the immunogenic 14-mer Hsp70 peptide TKD in the presence of pro-inflammatory cytokines, such as interleukin 2 (IL-2) [15] or IL-15. Hsp70-primed, CD3−/CD56+ NK cells expressing an elevated cell surface density of C-type lectin receptors, including CD94 and the activation marker CD69, efficiently recognize and kill membrane Hsp70-positive tumor cells [16,17] via granzyme B-mediated apoptosis [18]. 

Viable, membrane Hsp70-positive tumor cells have the capacity to actively release Hsp70 in extracellular lipid microvesicles with biophysical characteristics of exosomes [19]. Since exosomes are derived from normal cells, they do not present Hsp70 on their surface, and it is assumed that elevated exosomal Hsp70 levels in the circulation of tumor patients are predictive of the presence of membrane Hsp70-positive tumors. The compHsp70 sandwich ELISA which is based on two unique monoclonal antibodies-cmHsp70.1, cmHsp70.2- is able to measure both free and exosomal Hsp70 in the peripheral blood of tumor patients [20,21]. Circulating exosomal Hsp70 levels have been found to correlate with the viable tumor mass in mouse models [22]. Therefore, we speculate that the measurement of extracellular Hsp70 levels in the peripheral blood might improve tumor diagnosis and response monitoring in NSCLC patients because the blood can be collected more frequently by minimally invasive methods than multimodal tumor imaging approaches can be applied. 

Since Hsp70-primed CD3−/CD94+ NK cells play a critical role in recognizing and killing membrane Hsp70-positive tumor cells [15], the immunophenotype of NK cells and other peripheral blood lymphocyte subpopulations was comparatively assessed together with the circulating Hsp70 levels in patients with NSCLC in UICC stages IIIA, IIIB, and IV. In a previous study, circulating Hsp70 levels correlated with the gross tumor volume in NSCLC patients with squamous and adeno histology [23]. In the present study, elevated circulating Hsp70 levels were associated with an elevated prevalence of NK cells and higher tumor stages. We speculate that a significant drop in CD4+ T helper cells might be responsible for the impaired cytolytic activity of NK cells in advanced tumor stages when immunological tumor control has evidently failed, and the tumor load has increased. 

## 2. Materials and Methods

### 2.1. Study Participants

Blood samples were collected between 2015 and 2018 at the first diagnosis for the screening of their circulating Hsp70 status before the start of radio-chemotherapy in the Department of Radiation Oncology at the Klinikum rechts der Isar, TUM, as an inclusion criterium into the clinical phase II trial “Targeted Natural Killer Cell-Based Adoptive Immunotherapy for the Treatment of Patients with NSCLC after Radiochemotherapy: A Randomized Phase II Clinical Trial” [24]. 

The study was approved by the local Ethics Committee of the Medical Faculty of TUM. Written informed consent was obtained by all patients before blood collection. All procedures were undertaken in accordance with the Declaration of Helsinki, 1975/revised in 2008. 

### 2.2. Measurement of Free and Exosomal Hsp70 in Serum and Plasma Using the compHsp70 ELISA

Peripheral blood was collected by venipuncture. For serum preparation, the blood was collected in Serum Z/9 mL separator tubes (S-Monovette 7.5 mL, Sarstedt, Nürmbrecht, Germany) and was allowed to clot for 15 min at room temperature before being was centrifuged at 800× *g* for 10 min at room temperature. For plasma preparation, the blood was collected into EDTA KE/9 mL tubes and was centrifuged at 1500× *g* for 15 min at 4 °C. Aliquots of serum and plasma (300 µL) were stored at −80 °C. For the compHsp70 ELISA, Nunc MaxiSorb^TM^ flat-bottom 96 well plates (Thermo Scientific, Rochester, NY, USA) were coated overnight with 1 μg/mL cmHsp70.2 coating antibody (multimmune GmbH, Munich, Germany) in a sodium carbonate buffer (0.1 M sodium carbonate, 0.1 M sodium hydrogen carbonate, pH 9.6; Sigma-Aldrich). After washing, nonspecific binding was prevented by incubating the plates with a blocking solution (Liquid Plate Sealer^TM^, Candor Bioscience GmbH, Wangen i. Allgäu, Germany) for 30 min at room temperature. Following another washing step, the serum or plasma samples were diluted in a StabilZyme Select Stabilizer (dilution 1:5, Diarect GmbH, Freiburg i. Breisgau, Germany) and were added to the wells followed by another incubation period of 30 min at room temperature. An eight-point concentration standard curve using recombinant Hsp70 protein (0–100 ng/mL, multimmune GmbH, Munich, Germany) diluted in StabilZyme Select Stabilizer (Diarect GmbH, Freiburg i. Breisgau, Germany) was included in each assay. After another washing step, the plates were incubated with 200 ng/mL of biotinylated cmHsp70.1 monoclonal antibody (multimmune GmbH, Munich, Germany) in HRP-Protector (Candor Bioscience GmbH, Wangen i. Allgäu, Germany) for 30 min at room temperature in the dark. Following a final washing step, the plates were incubated with 57 ng/mL horseradish peroxidase (HRP)-conjugated streptavidin (Senova GmbH, Weimar, Germany) in HRP-Protector (Candor Bioscience GmbH, Wangen i. Allgäu, Germany) for 30 min at room temperature, after which the plates were washed and incubated with a substrate reagent (BioFX TMB Super Sensitive One Component HRP Microwell Substrate, Surmodics, Inc., Eden Prairie, MN, USA) for 15 min at room temperature. The colorimetric reaction was stopped by adding 2 N H_2_SO_4_, and an absorbance was read at 450 nm, corrected by the absorbance at 570 nm, in a microplate reader (VICTOR X4 Multilabel Plate Reader, PerkinElmer, Waltham, MA, USA). The comHsp70 ELISA kit is currently being developed as a commercial assay by DRG Instruments GmbH, Marburg, Germany [21]. 

### 2.3. Immunophenotyping of Lymphocyte Subpopulations by Multiparameter Flow Cytometry

The immunophenotyping of various lymphocyte subsets in the peripheral blood was performed by multicolor flow cytometry on a FACSCalibur™ flow cytometer (BD Biosciences, Heidelberg, Germany). For this, aliquots of the EDTA anticoagulated whole blood (100 µL) were incubated with different combinations of the following fluorescence-labeled antibodies: T cell antibodies; CD3-PerCP (BD-345766; Clone SK7), CD4-FITC (BD-555346; clone RPA-T4), CD8-FITC/PE (BD-347313; Clone SK1/BD-555366; clone RPA-T8), NK cell antibodies; CD56-FITC/APC (BD-345811; clone NCAM16.2/BD-555518; clone B159), CD16-PE (BD-555407, clone 3G8), CD94-FITC (BD-555888; clone HP-3D9), NKG2D-PE (FAB139P-R&D Systems; clone 149810), NKp30-PE (IM3709-Beckman Coulter; clone Z25), NKp46-PE (IM3711-Beckman Coulter; clone BAB281), and B cell antibody CD19-PE (555413-BD Biosciences; clone HIB19). All staining procedures included appropriate isotype- and fluorochrome-matched control antibodies.

After a 15 min incubation in the dark and washing with 2 mL of PBS/10% *v/v* FCS, the tubes were centrifugated at 500× *g* for 5 min at room temperature. To eliminate erythrocytes, the samples were incubated with BD FACS™ Lysing solution (1:9 dilution in ddH_2_O, 349202-BD Biosciences, 10 min) for 10 min in the dark at room temperature. After another wash, the step cells were analyzed on the FACSCalibur™. Lymphocytes were gated upon according to their FSC/SSC characteristics, and doublets were excluded. NK and T cell discrimination was based on the expression of CD3 and CD56, in combination with other surface markers detailed above.

For the analysis of regulatory T (Treg) cells, the cells were fixed by incubation with buffer A (1:10 in ddH_2_O, 51–9005451-BD Biosciences) for 10 min in the dark at room temperature. After two washing steps, the cells were permeabilized by incubation in Buffer C (1:50 in buffer A, BD-51-9005450). For gating CD3+ (CD3-PerCP, BD-345766), the T cells were divided into CD4+ (CD4-FITC, BD-555346) and CD8+ (CD8-FITC BD-347313) T cells. Then, the percentage of the CD25+ (CD25-APC, BD-340907) and FoxP3+ (FoxP3-PE, BD-560046) cells were determined within the CD4+ and CD8+ subpopulations. The percentage of positively stained cells was defined within a defined lymphocyte gate.

### 2.4. Multiplex Cytokine Analysis

A panel of different cytotoxicity markers and cytokines (granzyme B, IFN-γ, IL-2, IL-4, IL-6, IL-10) was measured in the blood of lung cancer patients using the MACSPlex Cytotoxic T/NK cell kit (Miltenyi Biotec B.V. & Co. KG, Bergisch Gladbach, Germany) according to the manufacturer`s recommendations.

### 2.5. Statistical Analysis

The comparison of the Hsp70 status of tumor patients versus the healthy control collective was achieved using an unpaired two-tailed Student’s t-test, whereas the differences across multiple groups were assessed using a one-way ANOVA and post hoc Tukey tests. Normal distribution was tested by the Shapiro–Wilk normality test; *p* values were considered statistically significant as follows: ns: not significant, * *p* < 0.05, ** *p* < 0.01, *** *p* < 0.001, **** *p* < 0.0001.

## 3. Results

### 3.1. NSCLC Patients in Different UICC Stages and Healthy Control Cohorts

A total of 113 patients with stage I and advanced NSCLC were included in the study (n = 113); 76 patients had an adeno (n = 76) and 37 a squamous cell carcinoma histology (n = 37). 6 patients were in UICC stage I (n = 6), 32 NSCLC patients were in UICC stage IIIA (n = 32), 33 were in IIIB (n = 33) and 42 were in IV (n = 42). All the patients were therapy naïve at the time of the blood draw. Patients with no proven NSCLC but with other tumor stages than those indicated, lung metastasis of different tumor origin, and small-cell lung cancer (SCLC) were excluded from the study. For the Hsp70 measurements, 42 healthy human volunteers (22 males/20 females, mean age 43 years, range 21–77, mean Hsp70 value 35.06 ng/mL) and 113 NSCLC patients (56 males/57 females, mean age 63 years, range 21–85 years, mean Hsp70 value 238.2 ng/mL), were included [21] (Figure 1). For the immunophenotyping of peripheral blood lymphocytes by flow cytometry, 16 healthy human volunteers were included (9 males/7 females, mean age 61 years, range 21–85 years). 

### 3.2. Comparison of Circulating Hsp70 Levels in NSCLC Patients in Different UICC Stages

Exosomal and free Hsp70 levels in the serum and plasma were measured with the compHsp70 ELISA and were significantly higher in NSCLC patients compared to the healthy control cohort (Figure 2A, *** *p* < 0.001), with NSCLC patients in stages IIIA (** *p* < 0.01) and IV (**** *p* < 0.0001) having significantly higher levels than the healthy controls (Figure 2B). Although Hsp70 levels in the NSCLC patients in stage IIIB were also elevated compared to the control group, the differences did not reach statistical significance. Circulating Hsp70 values in patients with UICC stage IV were significantly higher than those in patients with UICC stage IIIB (** *p* < 0.01) (Figure 2B). Hsp70 values in stage I NSCLC patients were lower than that of all the other stages but due to the relatively low numbers of patients (n = 6) in stage I, the differences reached were not statistically significant. We categorized the smokers into active smokers and ex-smokers. We defined the ex-smokers up to a maximum of 6 months before the study inclusion. A comparison of the Hsp70 levels in smokers (mean value 30 pack years) versus the ex-smokers (mean value 21 pack years) with low-grade tumors revealed no statistically significant differences (mean values 244.6 ng/mL versus 26.7 ng/mL, respectively). 

### 3.3. Immunophenotype in the Peripheral Blood of NSCLC Patients in Different UICC Stages

The presence and prevalence of the following lymphocyte subpopulations were determined in the peripheral blood of 113 patients with proven NSCLC at the first diagnosis by multiparameter flow cytometry: CD3−/CD19+ B cells, CD3+ T cells, CD3+/CD4+ helper T cells, CD3+/CD8+ cytotoxic T cells, CD3+/CD4+/CD25+/FoxP3+ regulatory CD4+ T (Treg) cells, CD3+/CD8+/CD25+/FoxP3+ regulatory CD8+ T (Treg) cells, CD3+/CD94+, CD3+/NKG2D+, CD3+/CD56+ NK-like T cells (NKT), CD56+/CD94+, CD3−/CD56+, CD3−/CD16+, CD3−/CD69+, CD3−/NKG2D+, CD3−/NKp30+, CD3−/NKp46+ NK cells. The gating strategy is outlined in the Appendix A.

No significant differences were detected in the composition of CD4 and CD8 regulatory T (Treg) cells and any of the NKT cell subpopulations in the IIIA, IIIB, and IV disease. Compared to the healthy controls, the prevalence of CD19+ B cells in stage IIIB patients and of CD4+ and regulatory CD8+ T (Treg) cells in stage IIIa, IIIB, and IV patients was significantly lower (Figure 3A,C,F). The proportion of CD3+ T cells was similar in all UICC stages and compared to the healthy controls (Figure 3B), whereas the proportion of CD4+ T helper cells was progressively lower in higher tumor stages, with a statistically significant value for stage IV (** *p* < 0.01, Figure 3C). In contrast, the prevalence of CD8+ cytotoxic T cells was significantly higher in stage IIIA (** *p* < 0.01) and IIIB diseases (* *p* < 0.05) compared to the healthy controls (Figure 3D).

The prevalence of CD3−/CD56+, CD3−/NKp46+, and CD3−/NKG2D+ NK cells significantly increased in NSCLC patients in UICC stage III to IV (* *p* < 0.05, Figure 4A–C). A significant increase in the proportion of CD3−/NKp30+ NK cells was observed between UICC stage IIIA and IIIB (* *p* < 0.05, Figure 4D). The proportion of CD3−/CD16+ NK cells was similar in all UICC stages, and there was no statistically significant difference in the prevalence of these NK subsets compared to the healthy controls. 

The percentage of CD3−/CD94+ and CD3−/CD69+ NK cells was found to be increased from UICC stages III to IV (Figure 4E,F). Compared to the healthy individuals, the prevalence of CD3−/CD94+ NK cells was significantly higher in stage I, stage IIIB, and stage IV (Figure 4E), and that of the CD3−/CD69+ NK cells progressively increased with UICC stages higher than III (Figure 4F).

The first results of a multiplex cytotoxicity and cytokine analysis (Multiplex Cytotoxic T/NK cell kit) of the blood of lung cancer patients with a low CD4+ T cell (mean 34.3%; n = 3) and a high CD3−/CD56+ NK cell (mean 16.6%) prevalence versus high CD4+ T cells (mean 61.3%; *n* = 2) revealed comparatively low IL-2 concentrations (mean 5.6 pg/mL versus 268.1 pg/mL, respectively), supporting the hypothesis that the limited effectiveness of NK cells could be due to the lack of the stimulatory cytokine IL-2, despite the presence of Hsp70 in the circulation (mean 303.3 ng/mL). As a consequence, the granzyme B (13.4 versus 69.1, respectively) levels in the liquid biopsies were also found to be reduced compared to patients with high CD4+ T cell ratios. In addition to IL-2, other stimulatory cytokines, such as IL-4, IL-6, and IFN-γ, appeared to be lower in patients with low CD4+ T cell ratios, whereas IL-10 levels appeared to be similar in both patient groups. 

## 4. Discussion

Since lung cancer is frequently only diagnosed at an advanced stage, the clinical outcome remains poor despite multimodal therapeutic approaches, including surgery, radiochemotherapy, targeted therapies, and immune checkpoint inhibitor blockade [1]. Consequently, there is a high unmet medical need for tumor biomarkers that are predictive for different UICC stages. The search for a broadly applicable tumor-specific biomarker has revealed Hsp70 as a promising candidate since Hsp70 is overexpressed in the cytosol of highly aggressive tumors of different entities, is presented on the cell surface of the tumor but not normal cells, and, importantly Hsp70 is actively released by tumor cells in extracellular vesicles with biophysical characteristics of exosomes [21]. Depending on its subcellular or extracellular localization, Hsp70 fulfills contradictory activities and acts as a “double-edged sword”. In the cytosol, Hsp70 assists with protein homeostasis and interferes with the apoptotic pathways, which in turn promotes tumor growth and mediates therapy resistance [11,12], whereas membrane-bound Hsp70 serves as a recognition structure for the immunocompetent effector cells. After cross-presentation, immunogenic peptides presented by different HSPs can stimulate CD8+ cytotoxic T cell-mediated and protective anti-cancer immunity. In the absence of immunogenic peptides, membrane-bound Hsp70 provides a recognition structure for Hsp70-primed NK cells [25]. In the previously-reported phase II clinical trial, we could demonstrate that patients with advanced membrane Hsp70-positive NSCLC could profit from ex vivo Hsp70-primed, autologous NK cells by exhibiting a prolonged progression-free survival. An increased prevalence of CD3−/CD94+ NK cells after adoptive NK cell transfer was associated with an improved clinical outcome [24]. In a pilot study, the overall survival of a patient with late-stage NSCLC could be extended by up to 32 months through combining Hsp70-primed NK cells with a second-line immune checkpoint inhibitor blockade addressing the PD-1 pathway [26].

The membrane-bound form of Hsp70 presented on viable tumor cells and exosomes, as well as free Hsp70, can be recognized by the unique cmHsp70.1 and cmHsp70.2 monoclonal antibodies [21,27]. Consequently, the compHsp70 sandwich ELISA, which is based on these two antibodies, measures both free and exosomal Hsp70 in circulation. Although free Hsp70 most likely originates from dying tumor cells, Hsp70 in extracellular lipid vesicles is actively released by viable tumor cells [19]. The amount of exosomal Hsp70 in the circulation of tumor patients is higher than that of free Hsp70. The expression density of membrane-bound Hsp70, which defines the amount of circulating Hsp70, can be increased by the therapeutic intervention (i.e., radiochemotherapy) and is higher in relapsed and metastatic tumors compared to therapy naïve primary tumors [13]. Therefore, the quantification of extracellular Hsp70 by the compHsp70 ELISA has the potential to determine not only the viable tumor mass but also to determine therapeutic responses [21].

In this study, we demonstrated significantly elevated Hsp70 levels in the circulation of NSCLC patients compared to healthy controls and higher circulating Hsp70 levels in advanced UICC stages. Previously, we have shown that elevated Hsp70 levels correlate with the gross tumor volume in NSCLC patients with adeno and squamous cell carcinoma histology [23]. Therefore, we assumed that Hsp70 might also serve as a tumor biomarker for advanced tumor stages with an unfavorable prognosis. Our findings are in line with the data of Zimmermann et al., who proposed Hsp27 and Hsp70 as potential cytosolic biomarkers to discriminate NSCLC patients in different tumor stages from healthy controls and COPD patients [28]. 

Since NK cells play a major role as the first line of defense against cancers, they have been used for adoptive cellular therapy strategies [29]. The ability to directly kill tumor cells without prior priming is crucial in early tumor control. After stimulation NK cells upregulate the expression of different activator NK receptor types, including the C-type lectin NKG2-receptors and the natural cytotoxicity receptors (NCRs), such as NKp30, NKp44, and NKp46, they all promote the cytolytic activity of NK cells. NK cell-mediated killing is mainly mediated by cytotoxic granules, such as perforin, granzymes, and granulysin, which are degranulated to initiate programmed target cell death or by death ligands, such as FasL and TRAIL, to further enhance target cell killing. The most recent findings suggest that NK cells are not only responsible for the non-inflammatory programmed cell death but also induce the immunological and more relevant inflammatory cell death, which triggers the release of “danger-associated molecular patterns” (DAMPs) and tumor antigens, which stimulate the adaptive immune system [30]. In contrast to CD8+ cytotoxic T cells that share the killing mechanisms mentioned above, NK cells are able to kill target cells via a unique mechanism termed antibody-dependent cellular cytotoxicity (ADCC), which is mediated by the low-affinity Fc gamma receptor CD16. Since our study demonstrates that the CD16-positivity was similar in all tumor stages, ADCC might not be stimulated by circulating Hsp70 in patients with advanced NSCLC.

Our group demonstrated that incubation of NK cells with the Hsp70 protein or the Hsp70-derived peptide “TKD”, together with the pro-inflammatory cytokine IL-2, stimulates their cytolytic activity against membrane Hsp70-positive tumor cells [15,31]. The elevated Hsp70 levels in the circulation of patients with advanced tumors and a higher tumor load might explain the increased prevalence of NK cells in these patients. It is especially worth mentioning that those cell surface receptors are significantly upregulated on NK cells (i.e., CD56, CD94) of NSCLC patients in advanced tumor stages, which are also increased after stimulation with Hsp70-peptide TKD and IL-2 [32]. The expression of the early activation marker CD69, which also controls the exhaustion of effector cells [33], progressively increased with advancing UICC tumor stages. However, the increased prevalence of CD3−/CD94+/CD69+ NK cells in the circulation of NSCLC patients in stages III and IV compared to the healthy controls did not result in improved tumor control. It was first speculated that this finding was due to an increase in the immunosuppressive regulatory CD4+ T (Treg) cells that are known to impair the effectiveness of CD8+ cytotoxic T cells, B cells, dendritic cells [34], and NK cells [35]. Notably, CD4+ regulatory T (Treg) cells can inhibit the T and NK cell function by suppressing the cytokine production of CD4+ T helper cells and by competing for pro-inflammatory cytokines. However, our study revealed a decreased prevalence of both CD4+ and CD8+ immunoregulatory T (Treg) cells in the circulation of advanced tumor stages. It remains to be determined whether these immunosuppressive cell types are more prevalent in close proximity to the tumor. 

CD4+ T helper cells fulfill multiple functions, such as licensing dendritic cells to prime CD8+ T cells, activating phagocytes, recruiting neutrophils, and stimulating the cytolytic function of immunocompetent effector cells, such as CD8+ T cells and NK cells, by releasing pro-inflammatory cytokines such as IL-2. Consequently, CD4+ T helper cells play a crucial role in tumor control as they enhance the activity of anti-tumoral effector cells [36]. The progressive decrease in CD4+ T helper cells with raising UICC stages, which might lower the IL-2 concentrations in the circulation, might also explain the insufficient tumor control by CD8+ T cells and the ineffectiveness of NK cells to kill membrane Hsp70-positive tumor cells, despite the presence of the danger signal Hsp70 in the circulation of patients with advanced UICC stages. The first results of a multiplex cytokine analysis that supports our hypothesis reveal lower IL-2, IL-4, IL-6, IFN-γ, and granzyme B concentrations in the circulation of lung cancer patients with low CD4+ T cells, high NK cell ratios, and high circulating Hsp70 levels compared to patients with high CD4+ T cell ratios. 

Taken together, Hsp70, which is increased in the circulation of NSCLC patients with advanced-stage tumors acts, as a danger signal for NK cell stimulation, and patients with advanced UICC stages exhibit an increased prevalence of NK cells that resemble the phenotype of NK cells after ex vivo stimulation with the Hsp70 peptide TKD and IL-2. We speculate that the insufficient tumor control, despite elevated CD8+ T cell and NK cell ratios, might be due to insufficient cytokine support of immunocompetent effector cells, which is induced by a reduced prevalence of CD4+ T helper cells. 

## 5. Conclusions

This study reveals that circulating Hsp70 levels could serve as a tumor biomarker for patients with NSCLC in advanced UICC stages. All patients in advanced tumor stages had significantly elevated Hsp70 levels in the circulation compared to the healthy control cohort, and Hsp70 levels progressively increased with higher UICC stages. These findings demonstrate the potential of Hsp70 measurements to predict an advanced tumor stage in NSCLC patients.

We have also demonstrated that the prevalence of CD3−/CD94+ NK cells and CD8+ cytotoxic T cells were greater in advanced tumor stages III and IV, whereas that of CD4+ T helper cells was decreased. We hypothesize that raised levels of circulating Hsp70 in higher tumor stages might support NK cell proliferation but that a lowered prevalence of CD4+ T helper cells tempers the capacity of cytolytic CD8+ T cells and NK cells to control tumor growth. 

## Figures and Tables

**Figure 1 cancers-14-05701-f001:**
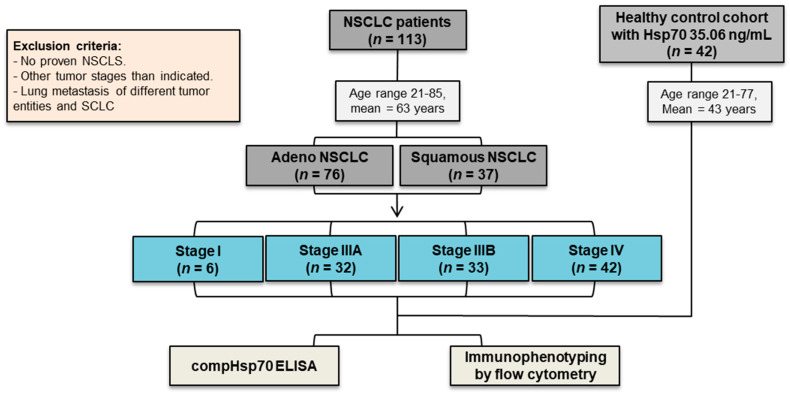
Schematic representation of the study. CONSORT diagram of the patient and healthy control cohorts.

**Figure 2 cancers-14-05701-f002:**
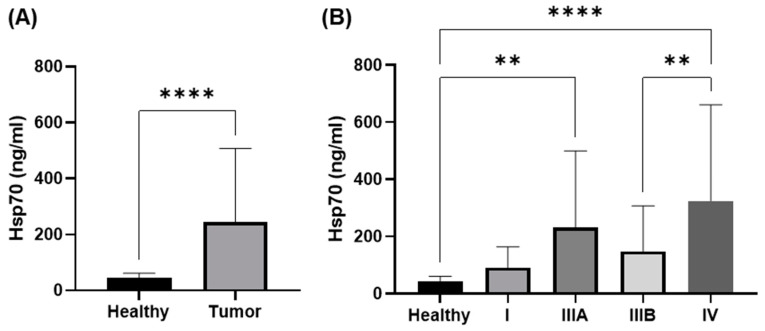
Measurement of exosomal and free Hsp70 in the peripheral blood of healthy volunteers vs. NSCLC patients as determined by the compHsp70 ELISA. The Hsp70 levels were compared in healthy controls (*n* = 42) vs. NSCLC patients in different UICC stages (*n* = 113) (**A**) and in healthy controls (*n* = 42) vs. NSCLC patients in UICC stages I (*n* = 6), IIIA (*n* = 32), IIIB (*n* = 33), and IV (*n* = 42) (**B**). Statistically significant differences were ** *p* < 0.01 and **** *p* < 0.0001.

**Figure 3 cancers-14-05701-f003:**
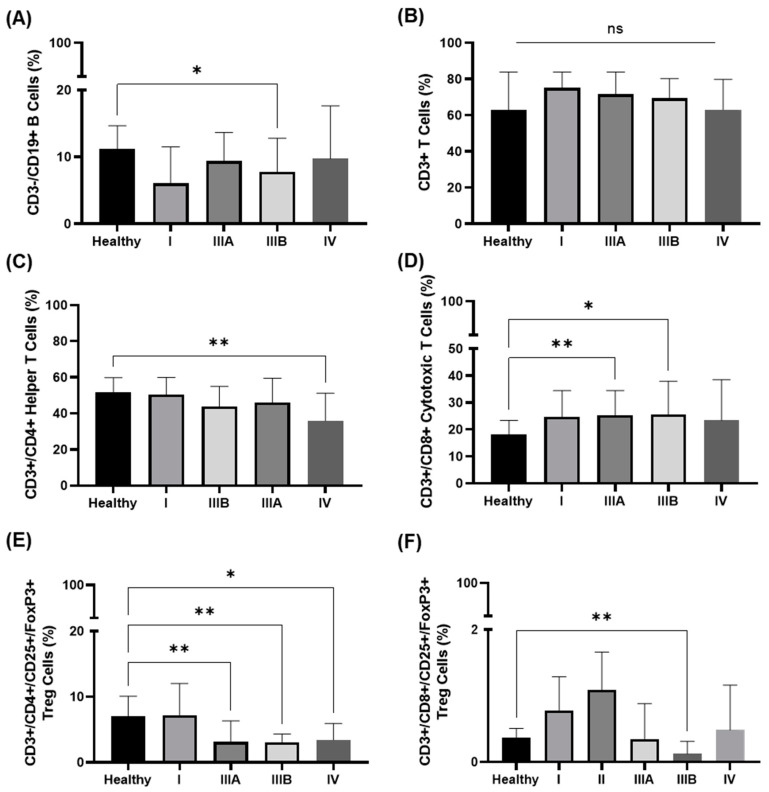
Frequency of B and T cell subsets in peripheral blood of healthy volunteers vs. NSCLC patients as determined by multiparameter flow cytometry. (**A**) The proportion of CD3+/CD19+ B cells in healthy volunteers is significantly higher than in NSCLC patients in UICC stage IIIB. (**B**) The proportion of CD3+ T cells in healthy volunteers is not significantly different to that of NSCLC patients in all different UICC stages. (**C**) With increasing UICC stages, the proportion of CD4+ T helper cells gradually decreases and reaches statistical significance in UICC stage IV. (**D**) With increasing UICC stages, the proportion of CD8+ T cytotoxic cells gradually increases and reaches statistical significance in UICC stages IIIA/B. (**E**) The proportion of CD4+ T regulatory cells in healthy volunteers are significantly higher than in NSCLC patients in all UICC stages (*n* = 107), apart from UICC stage I (*n* = 6). (**F**) The proportion of CD8+ T regulatory cells are significantly higher in healthy volunteers than in NSCLC patients in UICC stages IIIB. Statistically significant differences were * *p* < 0.05, ** *p* < 0.01.

**Figure 4 cancers-14-05701-f004:**
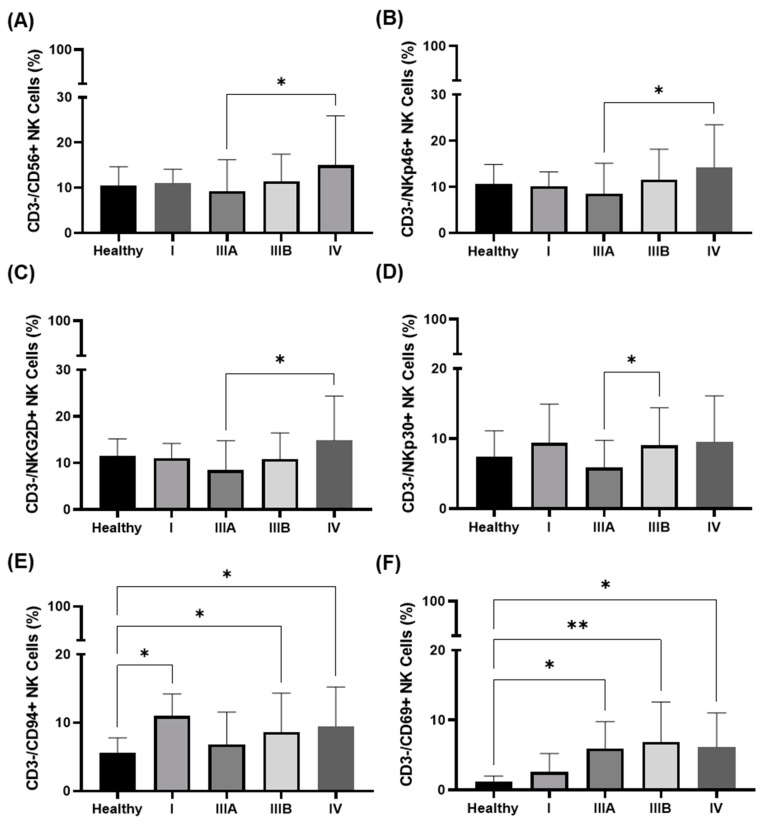
Frequency of NK cell subsets in the peripheral blood of healthy volunteers vs. NSCLC patients as determined by multiparameter flow cytometry. (**A**) The proportion of CD3−/CD56+, (**B**) CD3−/NKp46+, and (**C**) CD3−/NKG2D+ NK cells are significantly increased from UICC stage IIIA (*n* = 32) to IV (*n* = 42) (* *p* < 0.05). (**D**) The proportion of CD3−/NKp30+ NK cells is significantly increased from stage IIIA to IIIB. No significant differences were detected in these NK subsets in healthy volunteers vs. NSCLC patients in all different UICC stages (*n* = 113). (**E**) The proportion of CD3−/CD94+ NK cells are significantly increased in NSCLC patients in UICC stage IIIB compared to that of healthy volunteers. (**F**) The proportion of CD3−/CD69+ NK cells in NSCLC patients gradually and significantly increased compared to that of healthy volunteers. Statistically significant differences were * *p* < 0.05 and ** *p* < 0.01.

## Data Availability

The data presented in this study are available upon request from the corresponding author.

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
