# Peer review of "Elevated Levels of Circulating Hsp70 and an Increased Prevalence of CD94+/CD69+ NK Cells Is Predictive for Advanced Stage Non-Small Cell Lung Cancer"

_cancers, 2022, doi:10.3390/cancers14225701_

Round 1

Reviewer 1 Report

I thank the authors for this very important manuscript highlighting the lack of biomarkers for the management of patients with locally advanced or metastatic non small cell lung cancer. 

Circulating Hsp 70 and level of NK Cells have been presented as potential biomarkers of late stage. However, these data were obtained in comparison to healthy subjects. What about patients with early stage, stage I and II?

It would be important to have response data (ORR, PFS, OS) of patients with these biomarkers, can the authors provide them, to determine the clinical impact.  Do the responders have low Hsp70 levels?

When reading the results, the biomarkers that seem most interesting are the IL2 level and the CD4+ rate. What do the authors think?

Could we have the baseline characteristics of the population in a table.

In figure 2B, what is the significance difference between stage IIIA and IV, and between stage IIIA and IIIB please.

Author Response

Point-by-point letter

Reviewer 1

I thank the authors for this very important manuscript highlighting the lack of biomarkers for the management of patients with locally advanced or metastatic non-small cell lung cancer. 

  • Circulating Hsp70 and level of NK Cells have been presented as potential biomarkers of late stage. However, these data were obtained in comparison to healthy subjects. What about patients with early stage, stage I and II?

Answer

This point is well taken. Therefore, we included data of NSCLC patients in the early tumor stage I. However, the number of patients in this stage is limited since most NSCLC patients are diagnosed at a later tumor stage.

  • It would be important to have response data (ORR, PFS, OS) of patients with these biomarkers, can the authors provide them, to determine the clinical impact.  Do the responders have low Hsp70 levels?

Answer

This is also a helpful suggestion, however, for this study ethical approval has not been given to collect response data (ORR, PFS, OS). In an upcoming future study this aspect will be addressed.

  • When reading the results, the biomarkers that seem most interesting are the IL-2 level and the CD4+ rate. What do the authors think?

We completely agree with the reviewer. The drop in CD4 cells which seems to be associated with a decrease in pro-inflammatory cytokines such as IL-2 is most striking and might explain in part why patients with high CD8+ and NK cell levels do not control their tumors. The CD4 T cell drop was included as a key word.

  • Could we have the baseline characteristics of the population in a table.

Answer

The baseline characteristics of the patients are indicated in the CONSORT diagram and were included into the results part under 3.1.

Please note that the numbers of patients in the CONSORT diagram have changed since additional patients have been recruited into the study, including 6 patients in early stage I.

  • In figure 2B, what is the significance difference between stage IIIA and IV, and between stage IIIA and IIIB please.

Answer

The significance values between stage IIIa and IV have been included into Figure 2B.

Since additional patients have been included into the study the statistical significance was recalculated for all tumor stages including early stage patients as recommended by the reviewer. The results with more patients included remained the same.

The authors want to thank the reviewer for helpful suggestions and the editor for providing us with the opportunity to resubmit a revised version of the Ms.

Reviewer 2 Report

This is an interesting clinical study outlining the importance of circulating levels of HSP70 in NSCLC patients. In addition to the correlation of HSP70 levels in NSCLC patients, immune profile also shows correlation between clinical staging and different immune cell subtypes, supporting the prognostic role of HSP70 in NSCLC. I have few concerns which are outlined below.

1.       Have the authors tried to recruit chronic smokers without cancer as appropriate controls to check how smoking affect the HSP70 levels in blood in patients with and without lung tumor?

2.       Why were stages I and II NSCLC patients not included in the study? This can potentially indicate the role of HSP70 levels in early detection of NSCLC.

3.       Authors should provide the gating strategy for all the immune cell subsets

4.       Were the recruited patients treatment naïve at the time of blood draw?

Author Response

Point-by-point letter

Reviewer 2

This is an interesting clinical study outlining the importance of circulating levels of HSP70 in NSCLC patients. In addition to the correlation of HSP70 levels in NSCLC patients, immune profile also shows correlation between clinical staging and different immune cell subtypes, supporting the prognostic role of HSP70 in NSCLC. I have few concerns which are outlined below.

  1. Have the authors tried to recruit chronic smokers without cancer as appropriate controls to check how smoking affect the HSP70 levels in blood in patients with and without lung tumor?

Answer

We included Hsp70 values of chronic smokers versus ex smokers of tumor patients. Unfortunately we cannot provide data of non-smokers because all patients were smokers or ex smokers. The results are included into the text of the results part. Unfortunately data on the smoking behavior of the control group were not available. Since this aspect is interesting these data will be collected in a future study.

  1. Why were stages I and II NSCLC patients not included in the study? This can potentially indicate the role of HSP70 levels in early detection of NSCLC.

Answer

This point is well taken, additional data of NSCLC patients in stage I have been included into the study as recommended. However since most patients are diagnosed at a later stage the number of early stage patients was limited.

  1. Authors should provide the gating strategy for all the immune cell subsets

Answer

As recommended the gating strategy has been included into the Ms as a supplementary Figure 1.

  1. Were the recruited patients treatment naïve at the time of blood draw?

Answer

At the time of recruitment all patients were therapy naïve. This information has been included into the Results part.

The authors want to thank the reviewer for helpful suggestions and the editor for providing us with the opportunity to resubmit a revised version of the Ms.